# Macro-Nutritional Adaptive Strategies of Moose (*Alces alces*) Related to Population Density

**DOI:** 10.3390/ani10010073

**Published:** 2019-12-31

**Authors:** Yingjie Ma, Heng Bao, Roberta Bencini, David Raubenheimer, Hongliang Dou, Hui Liu, Sirui Wang, Guangshun Jiang

**Affiliations:** 1Feline Research Center of Chinese State Forestry Administration, College of Wildlife and Protected Areas, Northeast Forestry University, 26 Hexing Road, Harbin 150040, China; mayingjie328@gmail.com (Y.M.); baoheng@nefu.edu.cn (H.B.); wangsirui91@163.com (S.W.); 2Key Lab of Animal Ecology and Conservation Biology, Institute of Zoology, Chinese Academy of Sciences, 1-5 Beichenxi Road, Beijing 100101, China; 3University of Chinese Academy of Sciences, Beijing 100049, China; 4School of Agriculture and Environment, The University of Western Australia, 35 Stirling Highway, Perth 6009, Australia; roberta.bencini@uwa.edu.au; 5Charles Perkins Centre and School of Life and Environmental Sciences, The University of Sydney, Sydney, NSW 2006, Australia; david.raubenheimer@sydney.edu.au; 6College of Animal Science and Technology, Jinlin Agricultural University, Changchun 130118, China; douhongliang8@163.com; 7Institute of Tropical Agriculture and Forestry, Hainan University, No. 58, Renmin Avenue, Haikou 570228, China; liuhui_leen@163.com

**Keywords:** moose, nutritional strategy, N:C, forage quality, forage availability, nutritional geometry

## Abstract

**Simple Summary:**

Animals living in variable environments require flexible nutritional strategies for dealing with nutritional uncertainty. We investigated the diet and macro-nutritional strategies of male and female moose in six sites in northeast China, representing variable habitat quality and using spatially explicit capture-recapture to determine the local population density of moose during the snowy seasons. The moose populations experienced different forage availability and quality. Female and male moose equally tended to maintain a specifically balanced diet with a high ratio of protein and total nonstructural carbohydrates (N:C) across all populations, despite their differences in forage availability. A higher ratio of N:C in the vegetation was a positive indicator for population density.

**Abstract:**

The distribution area of moose in China has been shrinking back toward the north and northeast because of climate change and human disturbance, and the population number has been declining. Between 2011 and 2015, we studied moose at six sites in the northeast of China during the snowy seasons. We collected fecal samples and plant samples that were used to estimate population densities for moose, as well as their macro-nutrient selection. Out of a total of 257 fecal samples collected at six sites, we identified a total of 120 individual moose (57 females and 63 males). The population density (moose/km^2^ ± SE) was highest at Hanma with 0.305 ± 0.064 moose/km^2^ and lowest at Meitian with only 0.028 ± 0.013 moose/km^2^. Forage availability was different among sites, with the lowest availability at Mohe (58.17 number/20 m^2^) and highest was Zhanhe (250.44 number/20 m^2^). Moose at Zhanhe, Hanma, and Nanwenghe had a balanced diet with higher N:C (1:7), while at Meitian, Shuanghe and Mohe the N:C was 1:8. Our results indicate that the southern areas had low forage quality and quantity and this may be the reason for the distribution of the population of moose shrinking northward.

## 1. Introduction

Herbivores face many nutritional challenges, such as foods that vary in poorly digestible fiber [1], plant-produced toxins [2], and nutritionally imbalanced foods [3,4,5]. Therefore, it is important for herbivores foraging in heterogeneous environments to have a flexible feeding strategy to compensate for the natural variation in diet quality and quantity. It is also important to recognize whether and how herbivores can reach their nutritional requirements in the face of differences in the quality and quantity of diet on offer [6]. This is because they are the main factors regulating herbivore feeding patterns at the level of landscapes, as well as plant species and individual plant parts [7,8,9,10].

Protein and carbohydrates are the principal macro-nutrients that influence animal growth and reproduction, especially in folivores [5,11], with specific ratios of these nutrients often required for optimal performance [5,12,13]. However, searching for forages that contain protein and carbohydrates in optimal ratios is not an easy task for many wild animals [5,11]. This is especially true for herbivores because plants can be highly variable in their protein and carbohydrate content [14]. Additionally, access to high-quality foods may be restricted by the risk of predation [15], lack of forage availability [16], or lack of food diversity [17,18].

A species potentially exposed to such issues is the moose (*Alces alces*), the largest species of the *Cervidae* family, which lives in forest-wetland environments with a circumpolar distribution. In northeast China, moose are only found in the Greater Khingan Mountains and part of the Lesser Khingan Mountains, the southernmost edge of its distribution area in the world, where they have very few natural predators and hunting is strictly prohibited [19]. Moose are known to select areas with higher densities of mixed deciduous broad-leaf forest and mixed coniferous and broad leaf forest [20]. The distribution area of moose has been shrinking back toward the north and northeast, and the population number in China has been declining [21,22]. This decline is thought to be associated with climate change [22] and human disturbance [20], both of which fundamentally impact moose habitat and diet. Moose have different ways of adapting to heterogeneous environments, such as seasonal diet alterations [23], changes to metabolism [24], and variation not only in migratory strategies, but also in home ranges [25]. There is, however, limited knowledge on their macro-nutritional strategy at a regional scale and on how this herbivore adapts to heterogeneous environments at a local scale.

Macro-nutrient requirements, food selection, and dietary intake are likely to differ between sexes due to different physiological needs and post-ingestive nutrient processing [26,27,28]. Generally, metabolic rate decreases with increasing body mass [29]. Therefore, in ungulates where males are considerably larger than females, larger body sizes are associated with larger rumens and slower rate of passage of food through the gut [30]. Consequently, male ungulates are likely to be more efficient at extracting energy from fiber than females, and females need to compensate for this digestive constraint by either increasing foraging quantity or by selecting higher quality forage than that consumed by males [31].

To date, no studies have explored sex-specific macro-nutrient selection in moose. To examine this, we used proportion-based nutritional geometry, a multidimensional modeling approach [32,33,34], to explore the nutritional strategy of moose, and combined this with habitat quality to understand how the moose’s foraging strategy relates to its nutritional intake in heterogeneous environments. We expected that (1) under various circumstances, moose would choose to reach their optimal macro-nutrient targets as much as they can despite of the difference of forage availability, which means they would maintain a specific nutrient balanced diet [32,33]. Subsequently, we expected that (2) at sites where moose can maintain a better balanced diet (higher ratio of protein to carbohydrate), we would find a higher population density since nutritional intake condition of animals affects both survival and reproduction [35,36].

## 2. Materials and Methods

### 2.1. Study Area

The current distribution of moose in China is in the Greater Khingan Mountains and the Lesser Khingan Mountains in the northeast of China. For this research, we selected six areas in different geographical gradients (latitude 49°12′ N–53°18′ N, longitude 121°24′ E–125°30′ E) in the Greater and Lesser Khingan Mountains.

This study was conducted during the snowy seasons (December to March) in different years and areas in the northeast of China. It included six sites: Mohe (2011–2012, 2014–2015), Nanwenhe (2011–2012, 2014–2015), Zhanhe (2012, 2014–2015), Shuanghe (2011–2012), Hanma (2011–2012), and Meitian (2013–2014) (Figure 1). During the study, deciduous trees and shrubs were mostly leafless and snow cover (depth: 0–30 cm) was common throughout the winter. Temperatures during the study were −20–50 °C. We defined each site as ‘local scale’ and the whole six sites together as the ‘regional scale’.

In this region, moose are not migratory. Since the Chinese government strictly forbids the hunting of moose, we assumed that there would be no effect of hunting on moose population density.

### 2.2. Sample Collection

We used a total of 96 line transects, each ≥ 3 km long, systematically distributed at intervals of 2.5–3 km at each site, including 23 in Mohe, 17 in Nanwenghe, 20 in Zhanhe, 12 in Shuanghe, nine in Hanma, and 15 in Meitian. At each site, we also established 1316 survey plots (10 m × 10 m) at intervals of 200 m along the line transects. Five subplots (2 m × 2 m), one in the center and the other four at the corners, were also laid out in each 100-m^2^ plot to measure the number of annual new shoots of edible shrubs (heights: 0.5–3 m, Appendix A
Table A1), as well as the shoots browsed by moose, for a total of 6580 subplots [37]. We then used these plots to calculate the availability of forage. In total, we numbered 192,653 shoots, including 9502 in Mohe, 37,250 in Nanwenghe, 87,698 in Zhanhe, 19,578 in Shuanghe, 17,879 in Hanma, and 20,746 in Meitian.

To collect fecal and plant samples, we followed the tracks of moose in the snow that were fresh (<24 h), using backtracks for both researchers’ safety and to avoid disturbing the behavior of the animals. In total, we followed 84 snow tracks (>3 km) of moose, including 12 in Mohe, 17 in Nanwenghe, seven in Zhanhe, 16 in Shuanghe, 17 in Hanma, and 15 in Meitian. In total, we collected 257 fecal samples, including 39 in Hanma, 45 in Mohe, 47 in Nanwenghe, 44 in Shuanghe, 48 in Meitian, and 34 in Zhanhe.

### 2.3. Population Density

The population density of moose was estimated using spatially explicit capture-recapture [38]. This model is based on the hypothesis of closed population and on the activity of the species. Based on the individual identification results of the DNA analyses and our survey of GPS data for moose, spatially explicit capture-recapture was used to estimate the population density of moose in R Package (Package ‘SPACECAP’), using the results of individual identification to derive the spatial distribution points of all individuals in the research area. Since there is no research on the daily activity distance of moose in China, the calculation formula was based on research conducted in Europe [39]. According to that study, the activity distance of moose per hour is 0.075 km in winter. Thus, the daily activity distance of moose is 1.8 km.

### 2.4. Sex and Individual Determination

The fecal samples were collected from study sites and stored at −20 °C. Fecal DNA was extracted by QIAamp DNA Stool Mini Kit (QIAGEN, Hilden, Germany) following the manufacturer’s instructions. Multiplex PCRs were carried out with two pairs of primers, SRY12, designed to amplify the Y chromosome SRY region, and MAF46, designed to amplify one autosomal microsatellite locus as a positive control [40]. Each sample was amplified five times by independent parallel PCR. If more than three target bands appeared, the individual was judged to be male.

The MStools plug-in of Microsoft Office Excel software was used to judge whether the samples came from the same individual by the matching rate of multiple genotypes of each sample amplification result. Different samples came from same individual if: (1) all microsatellite loci had the same genotype; and (2) there were differences in one allele at only one locus. If more than four microsatellite loci failed to amplify, the sample was discarded for subsequent analysis.

### 2.5. Diet Composition of Moose

We collected moose feces and plant twigs from available plants foraged by moose at each site. Samples of moose feces were processed to identify the presence of macro-plant fragments using micro-histological analysis [37,41,42]. This methodology is widely used for estimating the diet composition of herbivores [43]. We followed similar methods in preparing reference slides (514 slides, check five lines of each, 2 cm × 5 cm) of plant species, which we used to identify plant fragments in fecal samples. Unidentified plant fragments were categorized as “unidentified” [44,45] and were not included in subsequent calculations. After identifying plant species, we estimated their relative frequency and converted them to the dry weight (DW%) of diets [45,46].

### 2.6. Nutritional Composition of Diet

Samples were ground and oven-dried at 65–70 °C for 48 h. Nutritional content was determined from the dry plant samples following Rothman et al. 2012 [47]. First, samples were ground in a Wiley Mill through a 1-mm screen. Crude fiber was measured via sequential analysis using an A2000i fiber analyzer (ANKOM Technology Corp., Macedon, NY, USA). Ash was measured by incinerating samples at 550 °C. Total nitrogen in plants was estimated using the Kjeldahl method (Kjeltec™ 8400, FOSS, Hillerød, Denmark). Crude protein was estimated by multiplying N% by 6.25 [48]. Crude fat was determined by Soxhlet extraction, in which samples were wrapped in filter paper and extracted by diethyl ether in a Soxhlet extractor at 70 °C for 4 h. Total nonstructural carbohydrates (TNC) were estimated by subtraction, where the sum of the percentages of fiber, crude fat, crude protein, and ash were subtracted from 100%. Nutritional composition of every fecal sample of moose was calculated by multiplying the specific dry weight (%) of each plant represented in the sample by its nutritional content. The nutritional composition of every individual was calculated as the mean of all fecal samples from the same individual. Similarly, nutritional composition of one site was calculated as the mean of the nutritional composition of all individuals found at that site.

### 2.7. Data Analyses

We used right-angled mixture triangle (RMT, proportions-based nutritional geometry) analysis [32] to examine the balance of macro-nutrients. We used a three-dimensional RMT, where each macro-nutrient was presented as percentage of total macro-nutrients (e.g., % protein = protein/(protein + fat + carbohydrate) × 100) on a dry matter basis (based on dry matter after the initial oven-drying). This method did not yield the composition of absolute proportion of separate nutrients. Rather, by comparing the relative percentage of nutrients we were able to focus on the macro-nutrients that we were interested in and exclude the effect of others [32]. We applied RMT at the regional scale (mean value of each population) to explore the differences between sites.

Margalef Richness Index (D), Shannon–Wiener Index (H’), species evenness index (J’), and species niche breadths (B) were calculated as:(1)D=(S−1)/ln
(2)$$H^{\prime }=-\overset{s}{\underset{i=1}{\sum }}pi\ln pi$$
(3)$$J^{\prime }=H^{\prime }/\ln S$$
(4)$$B=1/\sum pi^{2}$$
where *N* is the total detection number, *pi* is the proportion of individuals found in the *i*th species and *S* is the number of species in the sample [49].

We used multiple comparisons method (one-way ANOVA, Tamhane’s T2 test) to compare the difference of forage availability for moose across sites, and one-sample t-test to test the macro-nutrients differences in RMT. All tests were performed by SPSS v.22.0 (IBM corp., Armonk, NY, USA). We used linear regression to test the difference between the ratio of protein intake and total nonstructural carbohydrate intake (N:C) and population density, which was carried out with Graphpad Prism 5 (http://www.graphpad.com).

## 3. Results

Out of a total of 257 fecal samples collected at six sites, we identified a total of 120 individual moose, 57 females and 63 males, including eight females and four males at Mohe, 12 females and nine males at Nanwenghe, six females and 11 males at Zhanhe, six females and 12 males at Shuanghe, 19 females and 19 males at Hanma, and six females and eight males at Meitian.

The population density (moose/km^2^ ± SE) was highest at Hanma with 0.305 ± 0.064 moose/km^2^ and lowest at Meitian with only 0.028 ± 0.013 moose/km^2^ (Table 1).

### 3.1. Forage Availability and Diet Composition

Forage availability was lowest at Mohe (58.17 number of annual twigs/20 m^2^) and highest at Zhanhe (250.44 number/20 m^2^) (Figure 2). Except for among Meitian, Nanwenghe, and Zhanhe, as well as between Shuanghe and Hanma, there were significant differences (*p* < 0.05) between sites (Appendix A
Table A2).

The plant species detected from moose feces were birch (*Betula* spp.), *Tilia* spp., *Rhododendron* spp., *Lespedeza* spp., willow (*Salix* spp.), larch (*Larix* spp.), alder (*Alnus* spp.), aspen (*Populus* spp.), spruce (*Picea asperata* Mast.), elm (*Ulmus* spp.), hazelnut (*Corylus* spp.), pine (*Pinus sylvestris* Linn), and Mongolian oak (*Quercus* spp.). The moose staple food items varied across the six local sites (Appendix A
Figure A1). There were no significant differences between female and male moose in species richness, Shannon–Wiener, evenness, or niche breadth index. However, both female and male moose at Mohe had the lowest value of Shannon–Wiener, evenness, and niche breadth index. Female moose at Zhanhe had the highest value of Shannon–Wiener, evenness, and niche breadth index (Table 2).

### 3.2. Macro-Nutrient Balance of the Diet

At the local scale, according to the balance among protein (range from 9.72 to 11.33), fat (range from 12.02 to 13.81), and TNC (range from 75.39 to 77.54) of diet, female and male moose at Zhanhe consumed a diet with the highest protein and lowest fat (*p* < 0.05, Figure 3), while female moose at Shuanghe consumed a diet with the highest fat (*p* < 0.05, Figure 3). Both sexes of moose at Hanma consumed a significantly higher percentage of TNC and female moose had the highest protein, while male moose at Meitian consumed the lowest protein but the highest TNC (*p* < 0.05, Figure 3).

With respect to the ratio of N:C (range from 0.125 to 0.151), both female and male moose at Hanma, Zhanhe and Nanwenghe had higher N:C (1:7) than moose at Mohe, Shuanghe and Meitian (1:8). There was a pronounced positive linear relationship (R^2^ = 0.74, *p* = 0.028) between N:C and population density. With a higher N:C ratio, population density was higher (Figure 4).

## 4. Discussion

Our analysis indicated that moose functionally responded to differences in forage quality and quantity at different local sites. Although forage quality was different among sites, the balance between protein, fat, and TNC among them spanned within a narrow range, and this is consistent with our first hypothesis that moose would maintain a specific nutrient balanced diet. For ungulates, the winter period has severely limited forage availability compared with summer, and digestible protein is much lower in stems than leaves [50]. Although reproduction is often linked to the condition of moose in autumn [51], winter food was linked to fecundity and calf survival [52], and the amount of forage should be an important factor influencing foraging location [53,54]. Our results suggested, however, that forage availability was not a consistent predictor for better nutritional intake at the regional scale (larger scale) during winter [55,56,57]. At the local scale, separately, Meitian had a higher forage availability (Figure 2), but the moose at this site had low protein intake and N:C. On the other hand, Hanma had a low forage availability, however, the moose had a high protein intake and N:C. This suggests that forage availability was not critical in determining their nutritional intake balance. It was the forage quality and forage quantity together that gave them the chance to balance their diet during winter.

Our results demonstrated that higher ratios of N:C in the food were positively related to the population density, and this was in line with our second hypothesis that when moose can maintain a more balanced diet, they have a higher population density. Based on the nutrient balancing hypothesis [5,58], when sufficient food is available, the primary goal of an animal is to obtain a nutritionally balanced diet [47,59,60]. Felton et al. [61] demonstrated that in captivity moose were in line with the nutrient balancing hypothesis and reached a balance in macro-nutrients when they were provided access to sufficient nutrients. These authors also indicated that captive moose obtained a ratio of P:NPI (protein:non-protein intake) that included total nonstructural carbohydrate and fat of 0.12–0.41. In our study, the intake ratio of P:NPI by moose spanned a lower range of 0.11–0.13. Additionally, it was reported that moose require approximately 6.8% protein on a dry weight basis in their diet as a minimum requirement for maintenance [62], and our results showed that the percentage of protein ranged from 6.41% to 7.48%, indicating that moose in our study sites had limitation on protein intake in winter. Thus, the moose consumed winter forages that were near the limits of adequate protein content to support maintenance or reproductive requirements [63]. It was also confirmed that moose selected for plant material that matched a specific nutritional composition and they used birch foliage with more protein and less TNC [64].

Our results showed that both sexes of moose had a specific N:C ratio around 1:7, and when moose had a higher intake of protein, they had a higher ratio of N:C. It is reasonable that moose at our sites had no apparent nutritional segregation, since the forage quality of moose in winter was quite low. Similar results were reported for the greater kudu (*Tragelaphus strepsiceros*), in which females and males showed distinct separation in nutrient rich habitats, while there was no clear pattern of segregation in the poor habitats [65]. It is becoming more and more recognized that the consumption of macro-nutrients in optimal proportions is much more beneficial than the consumption of a particular macro-nutrient in an optimal amount [36,66].

Our results demonstrated that the population density of moose was positively correlated with the ratio of N:C, so higher ratios of N:C in the vegetation resulted in higher population density of moose in winter. Understanding the mechanisms governing the dietary choices of wild herbivores is fundamental to understand the adaptations of foragers, as well as their roles in structuring ecological communities [5]. Better quality food results in greater population densities of moose, suggesting that poor or reduced food quality could be one of the reasons for the decline in moose populations. Previous studies have demonstrated that the distribution of moose in China has been shrinking back toward the north and northeast [22]. However, our study indicated that forage quality and moose population density were better in Hanma and Nanwenghe (intermediate latitudes) and in the east site (Zhanhe) than other sites (Mohe, Shuanghe, and Meitian) in winter. In the future, we will need to explore the diets and nutritional balance of moose in summer.

## 5. Conclusions

Our results demonstrated that although moose experienced different forage quality and quantity under heterogeneous environments in winter, they tended to maintain a balanced diet (higher N:C) at a regional scale, and forage availability and quality should be put together to predict their nutritional state. When nutrients were severely restricted during winter, female and male moose did not adopt different foraging strategies due to nutritional restrictions. Additionally, a higher ratio of N:C was a positive indicator for population density. If climate change and human disturbance keep driving moose northward, the population size is likely to decline. The fact that southern areas had low forage quality and quantity may help to explain why the population of moose is shrinking northward.

## Figures and Tables

**Figure 1 animals-10-00073-f001:**
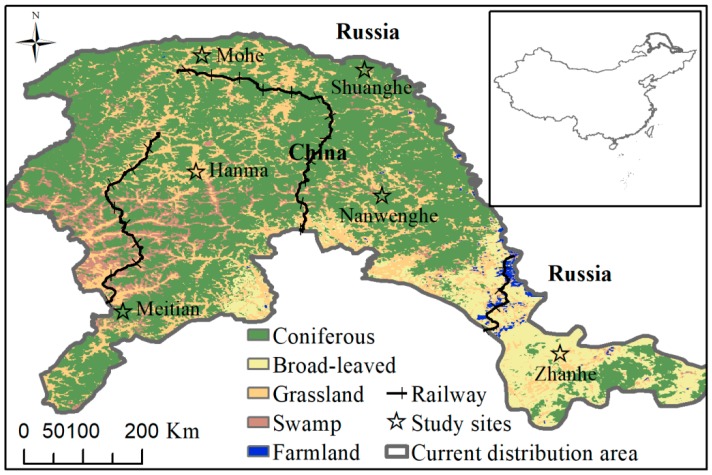
Location and vegetation of six sites where we studied the diet of moose (*Alces alces*) in northeastern China.

**Figure 2 animals-10-00073-f002:**
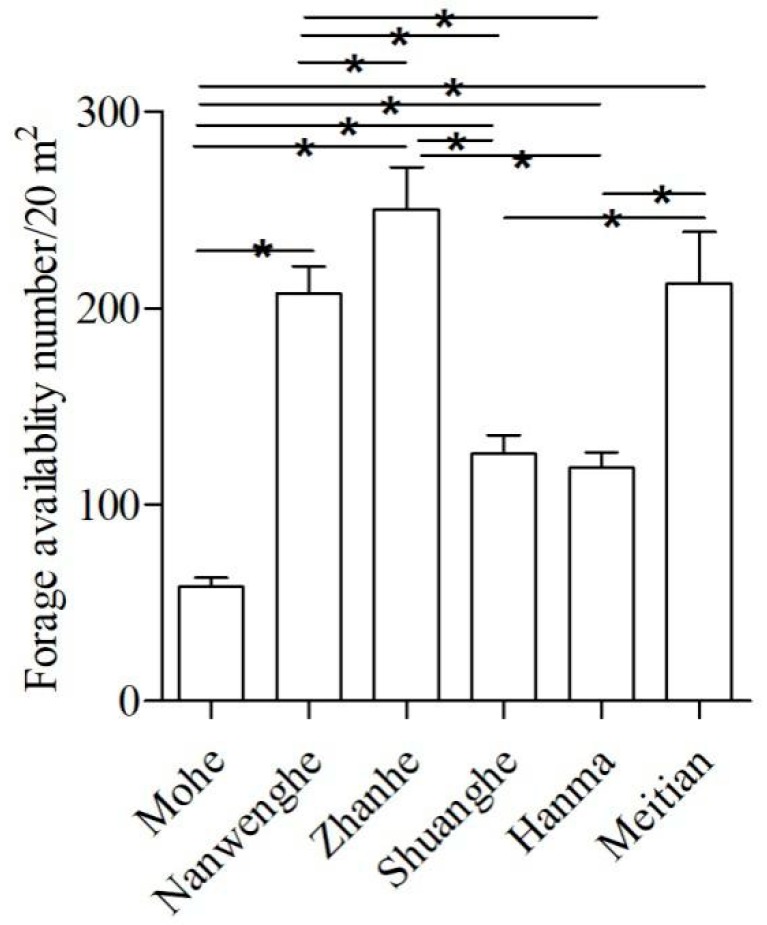
Forage availability expressed as the number of annual new shoots in the survey plots (10 m × 10 m) in Mohe, Nanwenghe, Zhanhe, Shuanghe, Hanma, and Meitian. Except for among Meitian, Nanwenghe, and Zhanhe, as well as between Shuanghe and Hanma, there were significant differences between sites. Asterisks indicate significant differences (F = 15.64, *p* < 0.05).

**Figure 3 animals-10-00073-f003:**
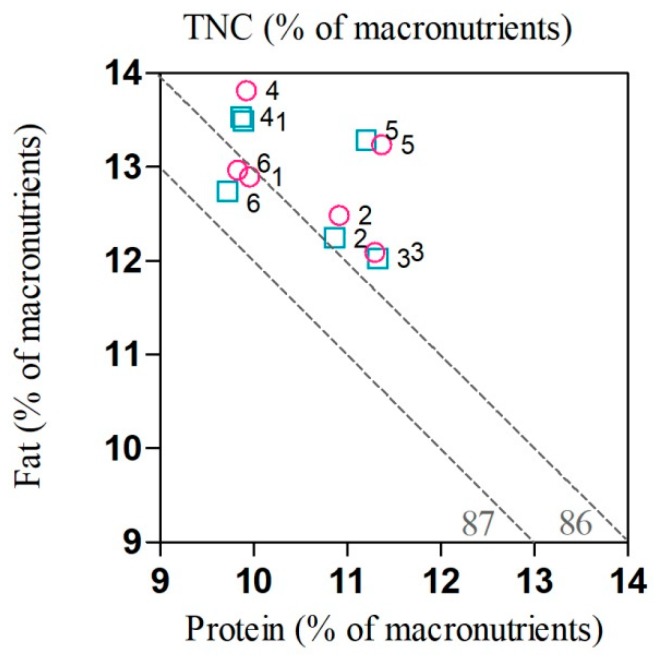
Right-angled mixture triangle showing the macro-nutrient balance in the diet of female (pink) and male (blue) moose at Mohe (**1**), Nanwenghe (**2**), Zhanhe (**3**), Shuanghe (**4**), Hanma (**5**), and Meitian (**6**). Macro-nutrients are expressed as percentage of total macro-nutrients (protein + fat + total nonstructural carbohydrates (TNC)), % of macro-nutrients. TNC is shown on the implicit *z*-axis, the value of which is inversely related with distance from the origin.

**Figure 4 animals-10-00073-f004:**
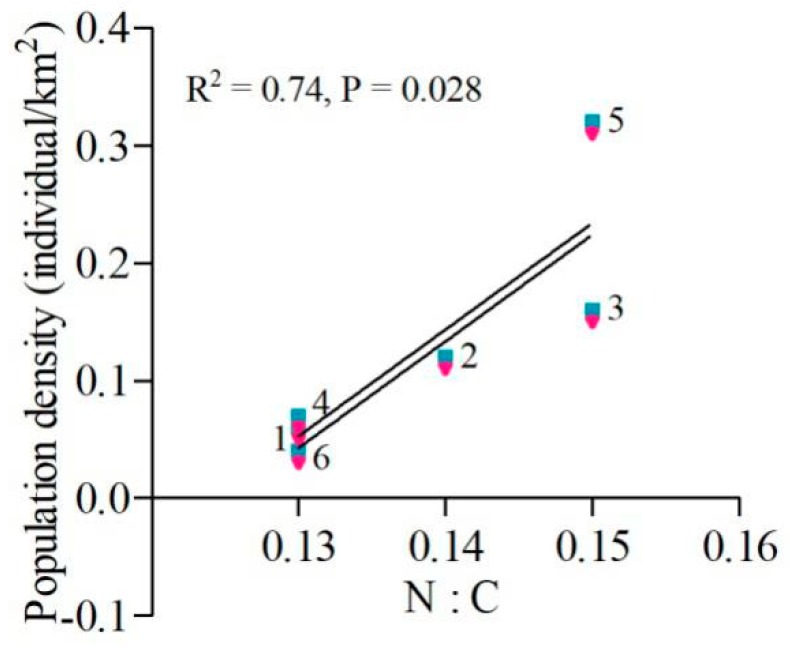
The linear relationship (solid line, R^2^ = 0.74, *p* = 0.028) between N:C (protein intake:TNC intake) and population density in both female (red) and male (blue) moose (male values were shifted +0.01 vertically to make them visible) at Mohe (**1**), Nanwenghe (**2**), Zhanhe (**3**), Shuanghe (**4**), Hanma (**5**), and Meitian (**6**).

**Table 1 animals-10-00073-t001:** Population density ± SE calculated with the spatially explicit capture recapture method in R.

Site	Population Density (moose/km^2^)	Area (km^2^)	Sampled Moose Individuals
Hanma	0.305 ± 0.064	216.3	38
Zhanhe	0.150 ± 0.051	156.44	17
Nanwenghe	0.111 ± 0.056	256.15	21
Shuanghe	0.056 ± 0.018	193.1	18
Mohe	0.052 ± 0.005	269.91	12
Meitian	0.028 ± 0.013	207.78	14

**Table 2 animals-10-00073-t002:** Food species, diversity, evenness indices, and niche width of moose during winter.

Site	Sex	Margalef *(S*)	Shannon-Wiener (*H’*)	Species Evenness (*J’*)	Species Niche Breadth (*B*)
Mohe	F	13	1.8	0.70	4.7
	M	12	1.76	0.71	4.63
Nanwenghe	F	13	2.11	0.82	6.98
	M	13	2.15	0.84	7.28
Zhanhe	F	13	2.33	0.91	8.33
	M	13	2.2	0.86	6.68
Shuanghe	F	13	2.26	0.88	8.21
	M	13	2.29	0.89	7.99
Hanma	F	12	2.14	0.86	7.11
	M	11	1.94	0.81	5.41
Meitian	F	11	2.09	0.87	7.38
	M	10	1.91	0.83	5.62

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
