# Peer review of "Macro-Nutritional Adaptive Strategies of Moose (Alces alces) Related to Population Density"

_animals, 2019, doi:10.3390/ani10010073_

Round 1
Reviewer 1 Report
Ma et al. – Macro-nutritional strategies of moose
Ma et al. investigate how the macro-nutrient selection of moose vary between male and female moose. Furthermore, they compare differences among sites with the expectation that moose will seek to maintain a specific nutritional balance and hence differences of habitat quality among sites will lead to differences in population density. They performed extensive field surveys to collect fecal samples and browsed twigs to be used in lab analyses to identify individual moose, plants consumed and the nutritional composition of diets. The authors found large differences in nutrient balance among sites, which correlated strongly with variation in population density among sites. The study provides additional insights into the foraging strategies of moose and complements the growing literature of herbivore foraging dynamics. I had some comments which I hope will improve the manuscript further which I have detailed below.
Major comments:
Study setup: An overarching concern I had was that the authors focused on fields surveys performed during winter. In general, the winter period has severely limited forage availability compared with summer, and for e.g. Schrempp et al. 2019 (https://dx.doi.org/10.1371%2Fjournal.pone.0219128) show how digestible protein is much lower in stems than leaves. That may explain why some sites had much lower than expected nutritional qualities. The nutrient balance in winter may be very important for over-winter survival, but reproduction is often linked to the condition of moose in autumn (and hence summer food), see for e.g. Testa & Adams 1992; https://doi.org/10.2307/1383026) although winter food may be linked to fecundity and calf survival (Allen et al. 2017; https://doi.org/10.1002/ece3.2594). I would therefore suggest highlighting these different aspects, so that you can link back to these in the discussion when discussing the relevance of your results in terms of nutritional qualities and moose population dynamics.
Minor comments:
Simple summary: There are several phrases in the summary which are not easily understandable for a non-expert in the field. Detailed mentions of methods (e.g. “proportions-based nutritional geometry”) and terms like “heterogeneous environments” should be more carefully explained, and at this stage a reader has no idea with N:C means.
Abstract: You could include some more information here regarding your study setup, e.g. collection of fecal samples and browsed twigs which were used to estimate not only population densities but also diet/nutritional compositions. You could also end your abstract with a broader relevance of your study (for e.g. in the introduction you emphasise the importance of climate change and human disturbance and how these impact habitat and diet).
L61 – L62: I find this sentence to be overly-simplifying of the existing literature. Several nutritional studies exist that link back to moose ecology, such as the Felton paper you cite and the reference therein. Furthermore, moose have different ways of adapting to heterogenous environment – such as seasonal diet alterations (Wam & Hjeljord 2010; https://doi.org/10.1007/s10344-010-0370-4), changes to metabolism (Regelin et al. 1985; https://www.jstor.org/stable/3801539) and variation in not only migratory strategies but also home ranges (Allen et al. 2016; https://doi.org/10.1002/ecs2.1524). I would suggest referencing to some of these known strategies of moose, and subsequently how your research complements this existing body of research.
L67 – L70: The cited article [28] states that surprisingly they found no difference in digested fibre between males and females despite the large body size of males, and instead behavioural differences may be important, for e.g. that females masticate forage much better than males. This therefore seems to contradict your expectation that females would need to select for higher quality forage or to consume more.
Study setup: Can you provide some info about the study area? How extreme is the climate (e.g. snow depths, temperatures). Are moose migratory in this region as is common in many other populations. You mention they are at the southern limit so perhaps not. Are the moose hunted at all and may this influence population densities?
L91 – L96: In considering new growth and available forage for moose, at what heights were plant measured? For example, moose tend to browse plants between 0.5m and 3m (Nichols et al. 2015; https://doi.org/10.1007/s00442-014-3196-z).
L96: please list (e.g. appendix) what you consider to be an edible shrub.
L97 – L104: Were all shoots collected from along fresh snow tracks? Since you followed 84 snow tracks, I am surprised that on average a moose browsed ~2300 shoots in such a short period of time. Were you able to assess the freshness of the bites? In my experience, moose use a limited home range in the winter months and do not move very far. You state you followed them for 3km though so I wonder how fresh the tracks remained as you backtracked the moose track?
L97 – L104: Do other ungulate species occur in the area? Could you distinguish bite marks from other species? For example Nichols et al. (2016; https://doi.org/10.1111/j.1755-0998.2012.03172.x) used eDNA procedures to identify different browsing species.
L109 – Please provide more information about the GPS data of moose that you refer to here. Is this your own GPS data or do you refer to other studies using GPS data?
L126 – Please indicate how you determined the moose’s daily activity distance? The distances travelled by individuals can have large impacts on SECR methods and subsequent density estimates, as shown recently in a study using camera trapping of moose (Pfeffer et al. 2018; https://doi.org/10.1002/rse2.67). Did you allow for any variation in daily travel distances amongst your study sites?
L184 – this is the first mention of your N:C ratio. Please provide a clear description of N and C, as neither letter is clearly linked to protein intake (N), nor total non-structural carbohydrate (TNC) = C.
Results
L187 – L189: Could you sex all fecal samples, and can you give the mean number of fecal samples per individual. This information is important for understanding the accuracy of the SECR analysis in generating the moose HRs, because it seems that on average you collected ~2 fecal samples per individual?
Figure 4 – Can you increase the detail of your N:C variable? The main results only refer to two ratios (1:7 and 1:8), whilst the graphs have been presented with the results rounded to two decimals places (i.e. only 0.13, 0.14, 0.15). Is it possible to display the results on a continuous scales? Figure 3 would suggest that you results have a level of accuracy to describe this in more detail.
Discussion: In general, I expected the authors to return to the topics mentioned in the introduction, for example the importance of understanding diet in heterogenous environments and how they results may be important for understanding the shrinking distribution of moose (e.g. climate change).
L246 – Is P : NPI the same as N: C? The values reported (0.11 – 0.13) appear to be the same but it becomes confusing when two different ratios are used to report the same result.
L247 – L253: Are moose foraging requirements the same in summer & winter? As mentioned above, they have the ability to adjust their metabolism, and as shown by Schrempp et al. 2019, stems have a much lower protein content that leaves, thus one may expect lower protein ratios in winter compared with summer.
Author Response
Ma et al. – Macro-nutritional strategies of moose
Ma et al. investigate how the macro-nutrient selection of moose vary between male and female moose. Furthermore, they compare differences among sites with the expectation that moose will seek to maintain a specific nutritional balance and hence differences of habitat quality among sites will lead to differences in population density. They performed extensive field surveys to collect fecal samples and browsed twigs to be used in lab analyses to identify individual moose, plants consumed and the nutritional composition of diets. The authors found large differences in nutrient balance among sites, which correlated strongly with variation in population density among sites. The study provides additional insights into the foraging strategies of moose and complements the growing literature of herbivore foraging dynamics. I had some comments which I hope will improve the manuscript further which I have detailed below.
Reply: Thanks for the recognition of the value our research. We have modified the text and figures according to your recommendations and comments. Below in bold are the point-by-point responses to your comments.
Major comments:
Study setup: An overarching concern I had was that the authors focused on fields surveys performed during winter. In general, the winter period has severely limited forage availability compared with summer, and for e.g. Schrempp et al. 2019 (https://dx.doi.org/10.1371%2Fjournal.pone.0219128) show how digestible protein is much lower in stems than leaves. That may explain why some sites had much lower than expected nutritional qualities. The nutrient balance in winter may be very important for over-winter survival, but reproduction is often linked to the condition of moose in autumn (and hence summer food), see for e.g. Testa & Adams 1992; https://doi.org/10.2307/1383026) although winter food may be linked to fecundity and calf survival (Allen et al. 2017; https://doi.org/10.1002/ece3.2594). I would therefore suggest highlighting these different aspects, so that you can link back to these in the discussion when discussing the relevance of your results in terms of nutritional qualities and moose population dynamics.
Reply: Thank you, we have included a thorough discussion of this issue. Please, find it in Discussion, lines 254-257.
Minor comments:
Simple summary: There are several phrases in the summary which are not easily understandable for a non-expert in the field. Detailed mentions of methods (e.g. “proportions-based nutritional geometry”) and terms like “heterogeneous environments” should be more carefully explained, and at this stage a reader has no idea with N:C means.
Reply: We changed them and made them clear in lines 21-29.
Our Simple Summary now reads as follows:
Animals living in variable environments require flexible nutritional strategies for dealing with nutritional uncertainty. We investigated the diet and macro-nutritional strategies of male and female moose in six sites in northeast China, representing variable habitat quality and used spatially explicit capture-recapture to determine the local population density of moose during the snowy seasons. The moose populations experienced different forage availability and quality. Female and male moose equally tended to maintain a specifically balanced diet with a high ratio of protein and total nonstructural carbohydrates (N:C) across all populations, despite their differences in forage availability. A higher ratio of N:C in the vegetation was a positive indicator for population density.
Abstract: You could include some more information here regarding your study setup, e.g. collection of fecal samples and browsed twigs which were used to estimate not only population densities but also diet/nutritional compositions. You could also end your abstract with a broader relevance of your study (for e.g. in the introduction you emphasise the importance of climate change and human disturbance and how these impact habitat and diet).
Reply: Thank you for your suggestion, we included further details in lines 30-42.
Our Abstract is now as follows:
The distribution area of moose in China has been shrinking back towards the north and northeast because of climate change and human disturbance, and the population number has been declining. Between 2011 and 2015, we studied moose at six sites of in the northeast of China during the snowy season. We collected fecal samples and plant samples that were used to estimate population densities for moose as well as their macro-nutrient selection. Out of a total of 257 fecal samples collected at six sites, we identified a total of 120 individual moose, 57 females and 63 males. The population density (moose/km2 ± SE) was highest at Hanma with 0.305 ± 0.064 moose/km2 and lowest at Meitian with only 0.028 ± 0.013 moose/km2. Forage availability was different among sites, the lowest one was Mohe (58.17 number/20m2) and highest one was Zhanhe (250.44 number/20m2). Moose at Zhanhe, Hanma and Nanwenghe had a balance diet with higher N:C (1:7), Meitian, Shuanghe and Mohe were 1:8. Our results indicate the southern areas had low forage quality and quantity and this may be the reason for the distribution of the population of moose shrinking northward.
L61 – L62: I find this sentence to be overly-simplifying of the existing literature. Several nutritional studies exist that link back to moose ecology, such as the Felton paper you cite and the reference therein. Furthermore, moose have different ways of adapting to heterogenous environment – such as seasonal diet alterations (Wam & Hjeljord 2010; https://doi.org/10.1007/s10344-010-0370-4), changes to metabolism (Regelin et al. 1985; https://www.jstor.org/stable/3801539) and variation in not only migratory strategies but also home ranges (Allen et al. 2016; https://doi.org/10.1002/ecs2.1524). I would suggest referencing to some of these known strategies of moose, and subsequently how your research complements this existing body of research.
Reply: Thanks, in our revised version we have referred to these known strategies of moose in lines 71-73.
L67 – L70: The cited article [28] states that surprisingly they found no difference in digested fibre between males and females despite the large body size of males, and instead behavioural differences may be important, for e.g. that females masticate forage much better than males. This therefore seems to contradict your expectation that females would need to select for higher quality forage or to consume more.
Reply: Apologies for quoting an incorrect reference. We have now changed this.
Study setup: Can you provide some info about the study area? How extreme is the climate (e.g. snow depths, temperatures). Are moose migratory in this region as is common in many other populations. You mention they are at the southern limit so perhaps not. Are the moose hunted at all and may this influence population densities?
Reply: We have included the information in the Method part in lines 95-108.
L91 – L96: In considering new growth and available forage for moose, at what heights were plant measured? For example, moose tend to browse plants between 0.5m and 3m (Nichols et al. 2015; https://doi.org/10.1007/s00442-014-3196-z).
Reply: We also recorded the browse heights of moose by following snow tracks; browse plants were measured as in Nichols et al. 2015.
L96: please list (e.g. appendix) what you consider to be an edible shrub.
Reply: We have added this as Appendix 3.
L97 – L104: Were all shoots collected from along fresh snow tracks? Since you followed 84 snow tracks, I am surprised that on average a moose browsed ~2300 shoots in such a short period of time. Were you able to assess the freshness of the bites? In my experience, moose use a limited home range in the winter months and do not move very far. You state you followed them for 3km though so I wonder how fresh the tracks remained as you backtracked the moose track?
Reply: We counted shoots to estimate forage availability and numbered them along the line transects. Fresh snow tracks (<24h) were used to collet fecal samples and browsed shoots.
L97 – L104: Do other ungulate species occur in the area? Could you distinguish bite marks from other species? For example Nichols et al. (2012; https://doi.org/10.1111/j.1755-0998.2012.03172.x) used eDNA procedures to identify different browsing species.
Reply: Yes, there are roe deer and other ungulates also occur in the same area. We followed the fresh tracks of moose, which are easily distinguished from those of other ungulates, and ensured that the bite marks were fresh. We also learnt what they eat with the micro-histological method. In another article we also compared the components of the diet of moose in winter from feces and field feeding signs: (Bao, H., Dou, H., Ma, Y., Liu, H., & Jiang, G. (2017). Moose winter diet components from feces and field feeding signs: consistency and variability related to forage availability and nutritional requirements. Ecological Research, 32, 1-8.)
L109 – Please provide more information about the GPS data of moose that you refer to here. Is this your own GPS data or do you refer to other studies using GPS data?
Reply: It was our own GPS data and we have now clarified this in line 132.
L126 – Please indicate how you determined the moose’s daily activity distance? The distances travelled by individuals can have large impacts on SECR methods and subsequent density estimates, as shown recently in a study using camera trapping of moose (Pfeffer et al. 2018; https://doi.org/10.1002/rse2.67). Did you allow for any variation in daily travel distances amongst your study sites?
Reply: Since there is no research on the daily activity distance of moose in China, the calculation formula was based on research conducted in Europe (Neumann W,Ericsson G,Dettki H,Radeloff VC. Behavioural response to infrastructure of wildlife adapted to natural disturbances.Landscape and Urban Planning,2013,114: 9-27). According to that study, the activity distance of moose per hour is 0.075km in winter, thus the daily activity distance of moose is 1.8km. We have added this in lines 135-138.
L184 – this is the first mention of your N:C ratio. Please provide a clear description of N and C, as neither letter is clearly linked to protein intake (N), nor total non-structural carbohydrate (TNC) = C.
Reply: Thank you, we add a clear description of it in lines 195-196 where we first mentioned it.
Results
L187 – L189: Could you sex all fecal samples, and can you give the mean number of fecal samples per individual. This information is important for understanding the accuracy of the SECR analysis in generating the moose HRs, because it seems that on average you collected ~2 fecal samples per individual?
Reply: We have now included how many fecal samples were used at each site in lines 199-202.
Figure 4 – Can you increase the detail of your N:C variable? The main results only refer to two ratios (1:7 and 1:8), whilst the graphs have been presented with the results rounded to two decimals places (i.e. only 0.13, 0.14, 0.15). Is it possible to display the results on a continuous scale? Figure 3 would suggest that your results have a level of accuracy to describe this in more detail.
Reply: As we were talking about the ratio of N:C, we prefer to present our data using this ratio. We have, however, added the continuous results too at line 242.
Discussion: In general, I expected the authors to return to the topics mentioned in the introduction, for example the importance of understanding diet in heterogenous environments and how their results may be important for understanding the shrinking distribution of moose (e.g. climate change).
Reply: We have now included this in the discussion, lines 294-302.
L246 – Is P : NPI the same as N: C? The values reported (0.11 – 0.13) appear to be the same but it becomes confusing when two different ratios are used to report the same result.
Reply: N:C is the ratio between protein and total nonstructural carbohydrate. P:NPI is the ratio between protein and non-protein intake, which includes fat. So they are different. This has been explained in lines 275-276.
L247 – L253: Are moose foraging requirements the same in summer & winter? As mentioned above, they have the ability to adjust their metabolism, and as shown by Schrempp et al. 2019, stems have much lower protein content that leaves, thus one may expect lower protein ratios in winter compared with summer.
Reply: It is going to be our next goal to find out moose summer foraging requirements, and comparing them with the winter ones. We have discussed this in discussion part in lines 301-302.

Reviewer 2 Report
This manuscript is using nutritional geometry to investigate the macronutritional strategies of male and female moose in six sites of northeast China. I think the manuscript is interesting but I think it needs some clarifications in methods and discussion. When reading the manuscript I was confused about which sample/data type was collected in which transect/plot and year. I was also surprised to see a low sample size in results after such a big monitoring effort. See comments below. I found the macronutritional approach very interesting but its connection with population density a bit confusing.
Methods
The data collection is not very clear to me. The sites were monitored for multiple years but temporal variation was not taken into account in any of your models? Also the final sample size seems very small to me: Page 5 line 189
“7 and 4 at Mohe, 11 and 9 at Nanwenghe, 6 and 11 at Zhanhe, 4 and 10 at Shuanghe, 10 and 10 at Hanma, and 6 and 8 at Meitian.”
Do these samples belong to different years?
I am a bit confused about which samples were collected during the line transects, the 1316 survey plots and the subplots. How are they connected? The way it is written it seems they were used to collect different types of samples?
Population density paragraph: is there a reference for the molecular biology and GPS data on moose? Is it part of this study? I guess you collected molecular data during you capture events and the GPS position of the capture site?
Results
How about the rest of the 257 samples? Could not be used/did not give anything identifiable?
Discussion
How can such low sample size be indicative of population density?
Minor comments
Page 1 line 18 – do you define “differences in quality and quantity of diet” an issue? Is it because you relate it to how human activities have changed the landscape? It is not clear, I suggest to rephrase.
Page 1 line 21 – “and combined” remove and
Page 2 line 42:45- I suggest to split this sentence.
Page 2 line 75 – “under various circumstances” do you mean in heterogeneous environment?
Author Response
Review 2
This manuscript is using nutritional geometry to investigate the macronutritional strategies of male and female moose in six sites of northeast China. I think the manuscript is interesting but I think it needs some clarifications in methods and discussion. When reading the manuscript I was confused about which sample/data type was collected in which transect/plot and year. I was also surprised to see a low sample size in results after such a big monitoring effort. See comments below. I found the macronutritional approach very interesting but its connection with population density a bit confusing.
Reply: Thank you for your comments, we have tried to clarify these points both in the methods and the discussion, as requested.
Methods
The data collection is not very clear to me. The sites were monitored for multiple years but temporal variation was not taken into account in any of your models? Also the final sample size seems very small to me: Page 5 line 189 “7 and 4 at Mohe, 11 and 9 at Nanwenghe, 6 and 11 at Zhanhe, 4 and 10 at Shuanghe, 10 and 10 at Hanma, and 6 and 8 at Meitian.” Do these samples belong to different years?
Reply: We collected these samples from different years, but finished each site collection in one year. This is because different sites are far from each other and it is difficult to collect them in a single year given the harsh winter conditions. For the sample size, we collected a total of 257 fresh fecal samples and determined their sex and individual resulting in around 15 individuals for each site, which is not a small size when you compare them to their population density.
I am a bit confused about which samples were collected during the line transects, the 1316 survey plots and the subplots. How are they connected? The way it is written it seems they were used to collect different types of samples?
Reply: The subplots were included in the plots, and these plots were laid out along the line transects that were used to estimate the forage availability.
Population density paragraph: is there a reference for the molecular biology and GPS data on moose? Is it part of this study? I guess you collected molecular data during your capture events and the GPS position of the capture site?
Reply: We collected both the molecular and GPS data as part of this work. The paragraph has been revised to make this clearer.
Results
How about the rest of the 257 samples? Could not be used/did not give anything identifiable?
Reply: 257 were the fecal samples that we collected to identify the sex as well as individual moose. Some of the samples collected belonged to the same individuals, so we identified only 120 individuals in total. This has been explained in the Methods.
Discussion
How can such low sample size be indicative of population density?
Reply: First of all, both of the investigation method and sample size meet the requirements of SECR density evaluation. Secondly, the individual numbers via fecal evaluation at each point were more than 15, which is not a small sample size.
Minor comments
Page 1 line 18 – do you define “differences in quality and quantity of diet” an issue? Is it because you relate it to how human activities have changed the landscape? It is not clear, I suggest to rephrase.
Reply: Thank you; we have rewritten the simple summary in lines 21-29.
Page 1 line 21 – “and combined” remove and
Reply: Thank you, we removed it.
Page 2 line 42:45- I suggest to split this sentence.
Reply: We have split this sentence as you suggested in lines 50-54.
Page 2 line 75 – “under various circumstances” do you mean in heterogeneous environment?
Reply: Yes.
